# Thermodynamic Coupling Simulation of CrN/Cr Composite Coating Barrel Bore

**Shaowei Wang** [1,*], **Chuanbin Wang** [1] and **Wenjun Li** [2]

1   State Key Laboratory of Advanced Technology for Materials Synthesis and Processing, Wuhan University of Technology, Wuhan 430070, China; wangcb@whut.edu.cn
2   Qingdao Kairui Electronics Co., Ltd., Qingdao 266000, China; liwenjun7190@163.com
*   Correspondence: wsw235042@whut.edu.cn

**Abstract:** The barrel is the core component of the artillery, and its inner coating is the key material to effectively protect the barrel and improve its service lifetime. Due to its good properties, CrN/Cr composite is a potential alternative to the currently used Cr coating. In this study, finite element simulation has been performed using the software Ansys Workbench to analyze the temperature field and coupled stress field of the Cr coating barrel and the CrN/Cr composite coating barrel, respectively, during the firing. The results showed that compared with Cr coating barrel, the CrN/Cr coating can reduce the temperature and significantly mitigate the stress in the coating/steel matrix interface; thus, it is expected the CrN/Cr coating can better protect the artillery barrel.

**Keywords:** CrN/Cr composite coating; barrel bore; temperature field; stress field; Workbench software

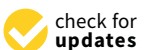

## 1. Introduction

As the decisive weapon in short-range combat, artillery undertakes a variety of tactical tasks, such as assault, air defense, anti-ship, land strike, and firepower suppression. It is widely used in many fields such as land, sea, and air [1–3]. The barrel is the core component of the artillery. It is prone to ablation and wear, which affects its maximum range, hitting accuracy, and service lifetime, because it is exposed to extreme conditions for a long time. Cr coating is now a common protective layer for the barrel of artillery, but it is hard and brittle, and its performance and processing technology have not been fundamentally improved for decades.

Compared with Cr, the CrN/Cr composite layer has a higher melting point and hardness [4], good wear resistance, and high temperature oxidation resistance. Moreover, its low expansion coefficient can mitigate thermal load and prolong the service lifetime. Although the manufacturing cost of CrN/Cr composite coating may be slightly higher than Cr, due to the better properties of CrN/Cr coating than Cr coating, it is expected that the lifetime of the barrel can be significantly prolonged, leading to an economic advantage of using CrN/Cr composite coating [4,5].

When the artillery is launched, the barrel and its surface coating are affected by the high temperature (instantaneous temperature as high as 2000–3000 °C) and high pressure (the maximum chamber pressure can reach 300–500 MPa) [3] caused by the combustion of gunpowder. In addition to that, during the launching process, the complex temperature field, the stress field, and the thermodynamic coupling effect exist. Thus, this study aims to elucidate these effects and provide a theoretical basis for the material design and performance optimization of the barrel.

Researchers have used traditional mathematical methods [6,7] and finite element methods [8] to analyze related artillery barrels, especially using Ansys, Abaqus, and other finite element software, focusing on the temperature field [9–11] or stress field of the Cr coating barrel [12,13]. However, finite element simulation of the promising CrN/Cr composite coating barrel has not been reported yet.

Thus, in this study, finite element simulation of the temperature field and thermomechanical coupling effect of the CrN/Cr composite coating barrel is performed, which aims to gain insight into the performance of this coating in extreme environment and provide theoretical guidance for the design of the artillery barrel. Since the artillery barrel is used in an environment involving the coupling stress of gunpowder gas flow, thermal stress, and chamber pressure, and Ansys Workbench is more capable of analyzing fluid and coupled fields; thus, it is adopted in this study. The temperature field and stress field of the barrel and the characteristic of interface between the composite coating and the gun steel matrix during launch are analyzed. The simulation of the traditional Cr coating barrel is also carried out to verify the feasibility of the present approach. The results from CrN/Cr composite coating barrel is compared with those from Cr coating barrel. We believe that the present work can provide a reference and guidance for the material design and performance optimization of the inner-bore coating of the artillery barrel.

## 2. Barrel Simulation Model

### 2.1. Theoretical Model of Finite Element Analysis

During the launching process of the artillery, the inner wall of the barrel will be subject to a transient impact. This impact includes the effects of heat and mechanics. In order to analyze the temperature field during the launching process, it is necessary to use the classical heat transfer theory as the basis to simulate the conduction of heat between different materials and analyze the distribution of the transient temperature field of different sections and nodes of the gun barrel with the launch time, so the classical heat transfer theory is the basis to simulate the temperature field of each position and node of the gun barrel [14].

The simulation of the temperature field follows the first law of thermodynamics, i.e., the law of conservation of energy:

$$Q - W = \Delta U + \Delta KE + \Delta PE \tag{1}$$

Among them, $Q$—heat, $W$—work, $\Delta U$—system internal energy, $\Delta KE$—system kinetic energy, and $\Delta PE$—system potential energy.

The heat transfer of the artillery barrel during the launch process usually does not consider the work, then

$$Q = \Delta U \tag{2}$$

In steady state analysis, $Q = \Delta U = 0$, in transient analysis, $q = \frac{dU}{dT}$, $q$ is the heat conduction rate of outflow or inflow.

In the finite element simulation of the temperature field of the artillery barrel, thermal convection is the most important form of exchange. Thermal convection refers to the exchange of heat between the solid surface and the surrounding fluid due to the temperature difference. Thermal convection is divided into natural convection and forced convection, which satisfies Fourier's law and Newton's cooling equation:

$$\frac{Q}{t} = \frac{KA(T_{hot} - T_{cold})}{d} \tag{3}$$

$$q = h(T_s - T_B) \tag{4}$$

Among them: $Q$—heat transfer in time $t$, $K$—heat transfer coefficient, $T$—temperature, $A$—heat transfer area, and $d$—heat transfer distance;

$h$—Heat convection heat transfer coefficient, $T_S$—the temperature of the solid surface, and $T_B$—the temperature of the surrounding fluid.

During the firing process, the gunpowder burns to produce high-temperature gas. Due to the great difference in temperature with the inner wall, it conducts forced heat convection exchange with the inner wall. The heat is transferred to the outer wall for natural heat conduction, and the outer wall conducts natural convection with the air.

Assuming that there is a temperature difference $\Delta T(x,y,z)$ inside an object, this difference will cause thermal expansion of the object. The amount of expansion can be expressed as $\alpha_T \Delta T(x,y,z)$; $\alpha_T$ is the thermal expansion coefficient, which expresses the thermal strain of the object. The physical equation will be:

$$\begin{cases} \varepsilon_{xx} = \frac{1}{E}\left[\delta_{xx} - \mu\left(\delta_{yy} + \delta_{zz}\right)\right] + \alpha_T \Delta T \\ \varepsilon_{yy} = \frac{1}{E}\left[\delta_{yy} - \mu\left(\delta_{xx} + \delta_{zz}\right)\right] + \alpha_T \Delta T \\ \varepsilon_{zz} = \frac{1}{E}\left[\delta_{zz} - \mu\left(\delta_{yy} + \delta_{xx}\right)\right] + \alpha_T \Delta T \\ \gamma_{xx} = \frac{1}{G}\tau_{xy}, \gamma_{yz} = \frac{1}{G}\tau_{yz}, \gamma_{zx} = \frac{1}{G}\tau_{zx} \end{cases} \tag{5}$$

In the formula, $\varepsilon_{xx}$, $\varepsilon_{yy}$, $\varepsilon_{zz}$—thermal strain component, $\delta_{xx}$, $\delta_{yy}$, $\delta_{zz}$, $\tau_{xy}$, $\tau_{yz}$, $\tau_{zx}$—stress component acting on the micro-element surface, where $\delta_{xx}$, $\delta_{yy}$, $\delta_{zz}$ represent normal stress, $\tau_{xy}$, $\tau_{yz}$, $\tau_{zx}$ represents the shear stress, E—Young's modulus, $G$—Shear modulus, and $\mu$—Poisson's ratio.

The finite element method can be used to solve above thermal stress equation, and the nodal displacement array of the element is set as:

$$q^e = \begin{bmatrix} \mu 1 & v1 & \omega 1 \cdots \mu_n & v_n & \omega_n \end{bmatrix} \tag{6}$$

Like the finite element analysis for a general elastic problem, the mechanical parameters in the element are expressed as the relationship between the displacement of the nodes, and they are

$$\mu^e = Nq^e \tag{7}$$

$$\varepsilon^e = Bq^e \tag{8}$$

$$\delta^e = D\delta^e - \delta^0 = DBq^e - D\varepsilon^0 = Sq^e - D\alpha_T \Delta T \begin{bmatrix} 1 & 1 & 1 & 0 & 0 & 0 \end{bmatrix}^T \tag{9}$$

Among them, $N$—shape function of unit, $B$—geometric matrix, $D$—elasticity matrix, and $S$—stress matrix. Substituting the displacement and strain equations of the element into the virtual work equation to eliminate the arbitrary nodal displacement variational increment $\delta q^e$, there is

$$K^e q^e = p^e + p_0^e \tag{10}$$

Among them:

$$K^e = \int_{\Omega^e} B^T DB d\Omega \tag{11}$$

$$P^e = \int_{\Omega^e} N^T \bar{b} d\Omega + \int_{\Omega^e} N^T \bar{p} dA \tag{12}$$

$$P_0^e = \int_{\Omega^e} B^T D\varepsilon^0 d\Omega \tag{13}$$

The above $P_0^e$ is also called the temperature equivalent load. Compared with the finite element formula for general elastic problems, the load end in the finite element equation increases the temperature equivalent load $P_0^e$.

### 2.2. Finite Element Model and Material Parameters

This article takes a gun barrel as the simulation object. The base material of the barrel is gun steel. The surface of the inner chamber is plated with Cr coating and CrN/Cr composite coating with the same total thickness. The thermodynamic physical property parameters of the material are shown in Table 1. Then, we ignore the fine structure of the barrel and the hydrogen produced during the launch of the gun, which does not renovate its mechanical characteristics [15–17]. We establish a three-dimensional finite element simulation model and perform a simplified analysis of the symmetry model, as shown in Figure 1.

**Table 1.** The thermodynamic parameters of gun barrel and coating materials.

| – | Density (g/cm³) | Coefficient of Thermal Expansion (10⁻⁶/K) | Thermal Conductivity (W/m·K) | Specific Heat Capacity (J/kg·K) | Elastic Modulus (GPa) | Poisson's Ratio |
|---|---|---|---|---|---|---|
| CrN | 6.14 | 5.2 | 11.7 | 850 | 402 | 0.30 |
| Cr | 7.19 | 9.4 | 83.6 | 505 | 200 | 0.12 |
| Gun steel | 7.80 | 12.1 | 40.8 | 460 | 207 | 0.29 |

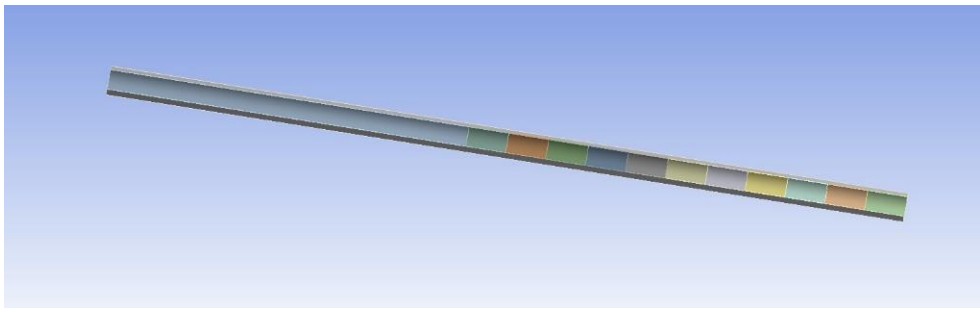

**Figure 1.** The finite element simulation model of the gun barrel.

Figure 2 shows the number of grid divisions and cross-sectional diagrams. The number of nodes is 374,856 and the number of grids is 73,200. The overall grid quality is high.

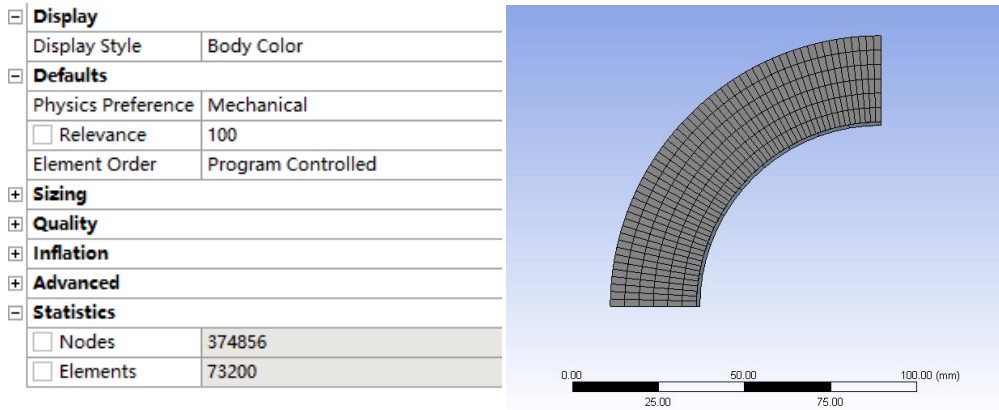

**Figure 2.** Diagram of cross-section grid.

### 2.3. Initial and Boundary Conditions

#### 2.3.1. Initial Conditions

Before the gun is fired, the temperature on the surface of the inner barrel of the barrel is room temperature, and the pressure on the inner barrel is one atmosphere [8].

#### 2.3.2. Boundary Conditions

When the artillery is launched, the high-temperature gas produced by the combustion of the gunpowder transfers heat to the barrel of the artillery, causing the temperature of the inner chamber to rise sharply, and thermal convection is the main form of heat exchange [13], and it is forced convection heat exchange. According to the interior ballistic equation, the gunpowder gas temperature as a function of time can be obtained [8]; the heat transfer coefficient is calculated according to the similarity criterion equation, and the average convective heat transfer coefficient of the gunpowder gas can be obtained as a function of launch time [18]; according to the internal ballistic equation, the dynamic changes of time and displacement can be taken into account, and the pressure of the

gunpowder gas can be obtained as the launch time. These curves are shown in Figure 3. In this study, the stress we analyzed is a coupled stress [19].

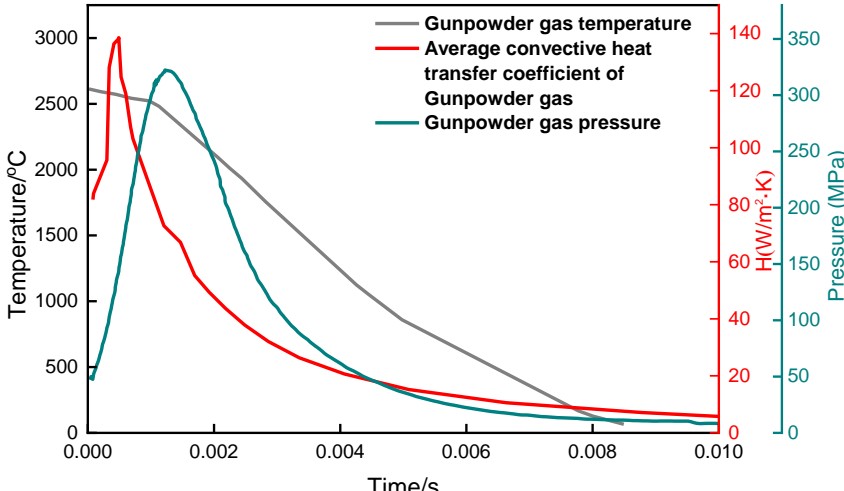

**Figure 3.** Variation curves of gunpowder gas temperature, average convective heat transfer coefficient and gas pressure with gun firing time.

## 3. Simulation Analysis of the Temperature Field of the Barrel

Figure 4 shows the variation curves of the maximum temperature of the inner chamber of the Cr coating barrel and CrN/Cr composite coating barrel with the firing time. The illustration in Figure 4 is a partial enlargement and comparison with the results reported in the literature. The authors of [3] use Abaqus finite element software, and this article uses Workbench finite element software to perform simulation calculations under the same initial conditions and boundary conditions. It can be seen that the calculation results are similar. The maximum temperatures are 529 and 513 °C, respectively, and the time to reach the maximum temperature is almost the same. This shows that the finite element method used in this article has a high accuracy.

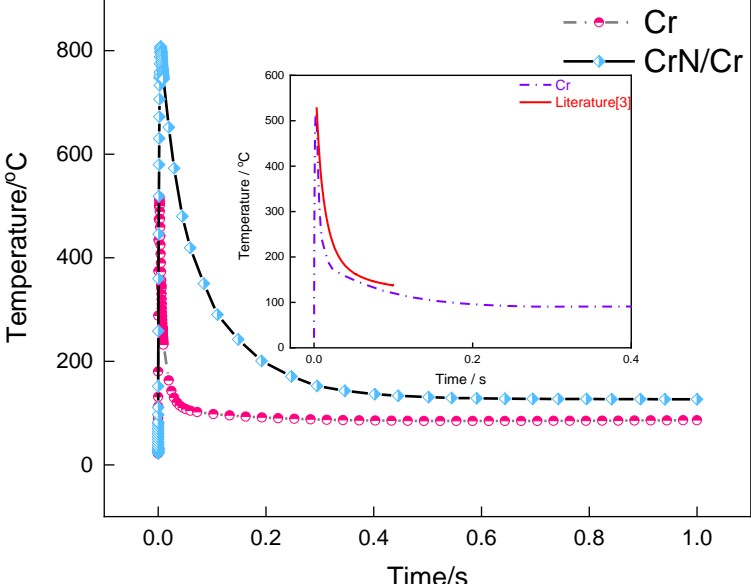

**Figure 4.** Variation curves of maximum temperature in bore of Cr coating barrel and CrN/Cr composite coating barrel with gun firing time.

It can be seen from Figure 5 that, regardless of the Cr coating barrel or a CrN/Cr composite coating barrel, when the gun is fired, the temperature of its inner chamber first rises sharply, then drops rapidly, and finally stabilizes. It reaches the highest temperature in an instant, and the duration of high temperature is also extremely short.

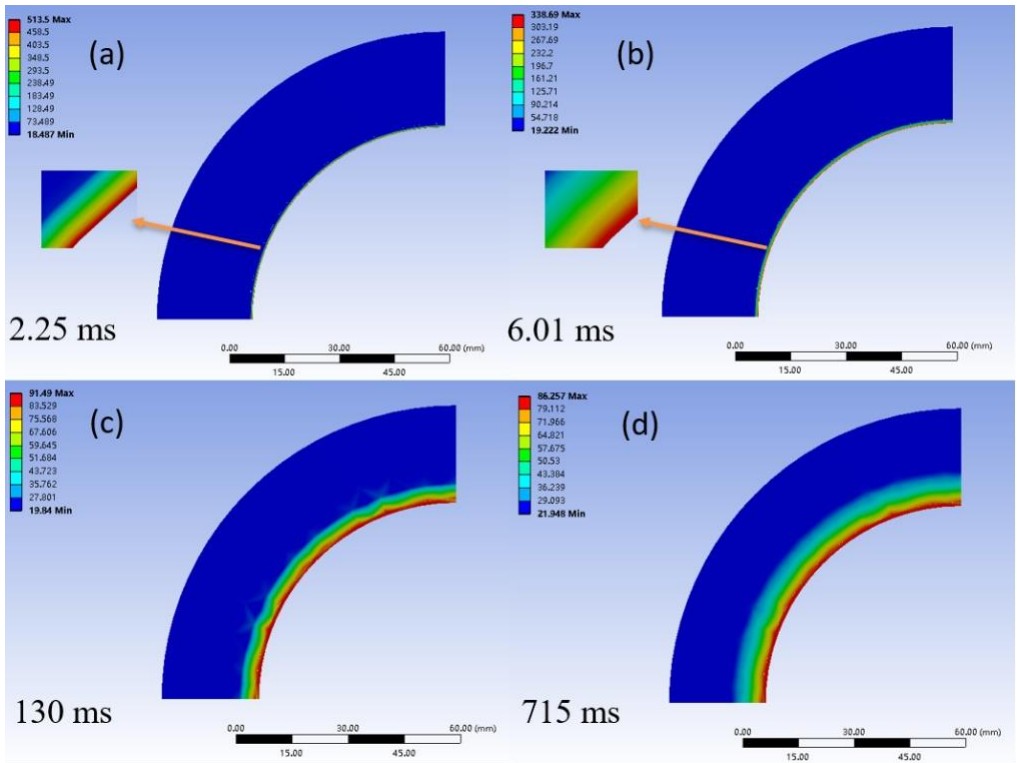

**Figure 5.** Cloud diagrams of the temperature of the A section (Cr coating). (**a**) 2.25 ms, (**b**) 6.01 ms (**c**) 130 ms and (**d**) 715 ms after the gun is fired.

Compared with the Cr coating barrel, the peak temperature of the CrN/Cr composite coating barrel is higher, and the decrease rate is gentler. This is because the thermal conductivity of CrN (11.7 W/m·K) is much smaller than that of Cr (83.6 W/m·K). Under the same boundary conditions, due to the forced convection exchange between the high-temperature gas of the propellant and the inner chamber coating, the surface heat of the CrN/Cr composite coating is difficult to transfer to the inner Cr coating and the gun steel matrix, leading to the accumulation of heat, and consequently, resulting in higher temperatures. The section with the most severe temperature change is selected as the A section. Figures 5 and 6 show the cloud diagrams of the temperature of the A section of the Cr coating barrel and the CrN/Cr composite coating barrel at different time, respectively. Figure 5a corresponds to the time that the maximum temperature of the Cr coating barrel was reached. The heat was transferred to the steel substrate at 6.01 ms (Figure 5b). The temperature of the barrel tends to be stable at 130 ms (Figure 5c) and later (Figure 5d). The diagram that CrN/Cr composite coating barrel reached the maximum temperature is presented in Figure 6a and its corresponding diagrams at 6.01, 130, and 715 ms are shown in Figure 6b–d in order to be compared with its counterparts in Figure 5.

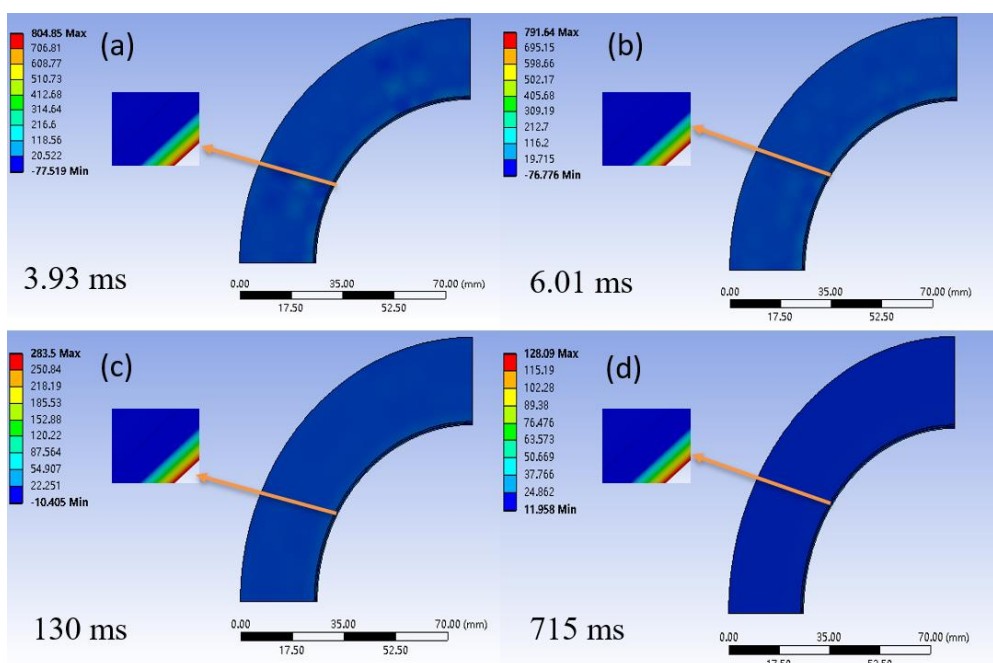

**Figure 6.** Cloud diagrams of the temperature of the A section (CrN/Cr composite coating). (**a**) 2.25 ms, (**b**) 6.01 ms, (**c**) 130 ms and (**d**) 715 ms after the gun is fired.

In Figures 5 and 6, it can be found that the time that the CrN/Cr composite coating barrel reaches the maximum temperature is 3.93 ms, which is slightly later than the time that the Cr coating barrel reaches the maximum temperature. At the same moment, the area where the temperature of the Cr coating barrel changes significantly, and the position within 12 mm from the inner wall changes more obviously, while the CrN/Cr composite coating barrel only changes within 0.1 mm of the inner wall. This is because the thermal conductivity of CrN is lower than that of Cr, but its specific heat capacity is higher than that of Cr (Table 1), which makes the CrN layer bear a greater amount of heat, and the heat of the inner wall could not transmit to the steel substrate in time, leading to greater thermal stress [3].

Figure 7 shows the maximum temperature of the bonding interface between the CrN/Cr composite coating and the steel matrix as a function of the gun firing time. The maximum temperature curve of the interface between Cr coating and the steel matrix is superimposed for comparison. The illustration in Figure 7 is a partial enlarged view up to 0.05 s. For the Cr coating, the interface temperature reaches the peak within 0.1 s and then slowly decreases. The peak temperature is about 100 °C, while, for the CrN/Cr composite coating, the interface temperature rose to 54 °C in a shorter period of time, followed by a sharp decline. This is due to the fact that the surface CrN has a higher specific heat capacity (Table 1), which results in a lower temperature at the interfaces between the CrN/Cr composite coating and the steel matrix. Meanwhile, because of the poor thermal conductivity and high specific heat capacity of CrN, the temperature of the interface between the CrN and Cr layer is lower than the temperature at the middle of the pure Cr coating. Thus, for the CrN/Cr composite coating barrel, the temperature difference between the Cr layer and the steel matrix is smaller than that of the pure Cr coating barrel; as a consequence, the CrN/Cr composite coating/matrix interface reached the peak temperature in a shorter time. It is safe to conclude that the CrN/Cr composite coating has a better heat insulation effect for the steel matrix of the gun barrel.

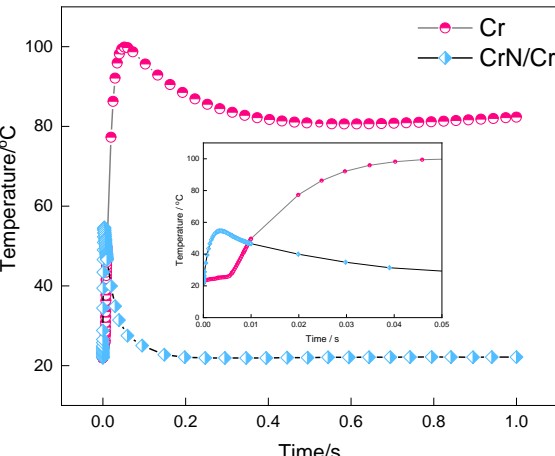

**Figure 7.** Variation curves of maximum temperature of bonding interface between Cr coating and CrN/Cr composite coating and gun steel matrix with gun firing time.

## 4. Simulation of the Stress Field of the Barrel

Figure 8 shows the variation curve of the maximum stress in the inner chamber of the Cr coating barrel and CrN/Cr composite coating barrel with the firing time of the gun. It can be seen that the inner bore stress of the two coating barrels increases first and then decreases slightly. This is the result of the coupling of thermal stress generated by the propellant gas pressure and the inner bore temperature field. As can be seen from Figure 5, the gunpowder gas pressure first increases sharply and then decreases rapidly with the firing time of the gun, and the barrel stress generated by the gunpowder gas pressure also shows a trend of first increasing and then decreasing; at the same time, the high temperature gunpowder gas continues to be on the body. The internal bore of the tube carries out forced heat exchange, so that the temperature of the internal bore of the body tube continues to rise, and the resulting thermal stress is also increasing. At the moment the gun is fired, the pressure of the gunpowder gas is very large, which plays a major role in the inner chamber stress; then, as the gas pressure drops and the inner chamber temperature rises, the thermal stress will play a leading role, so the inner chamber stress appears as shown in Figure 8. On the other hand, for the CrN/Cr composite coating barrel, due to the higher temperature in the inner chamber during launch (Figure 5), it has greater thermal stress, so the peak stress (1983 MPa) is also higher than the Cr coating barrel (1160 MPa).

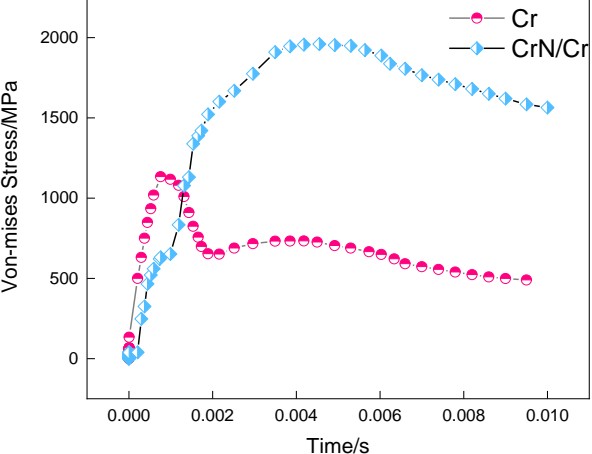

**Figure 8.** Variation curves of maximum stress in inner bore of Cr coating barrel and CrN/Cr composite coating barrel with gun firing time.

Figures 9 and 10 show the total deformation of the Cr coating barrel and the CrN/Cr composite coating barrel under the coupling effect of thermal stress and chamber pressure. It can be seen that the maximum deformation of the Cr coating barrel during launch time is 0.56 mm, and the maximum deformation of the CrN/Cr composite coating barrel is 0.29 mm, which is about half of the Cr coating barrel, attributing to higher elastic modulus of CrN/Cr than Cr (Table 1), resulting in a smaller deformation of the barrel. The maximum deformations of the two coatings appear at a position about 900 mm away from the bottom of the bore, which indicates that this position should be more importantly considered when coating the barrel with CrN/Cr composite coating to prevent cracks, peeling, etc.

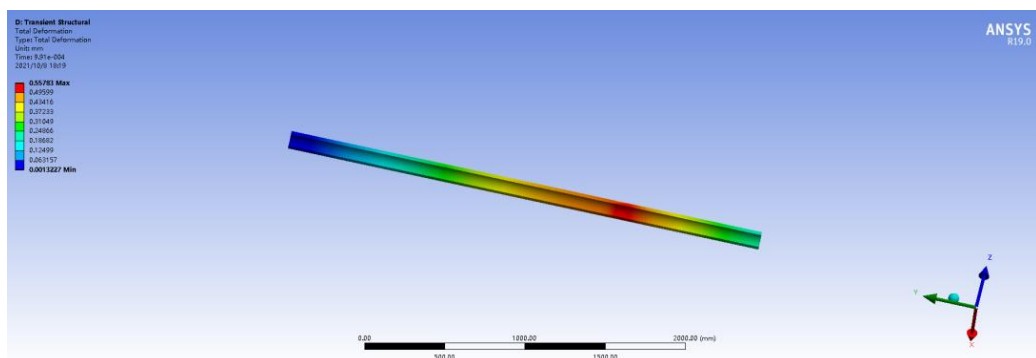

**Figure 9.** Total deformation of Cr coating barrel.

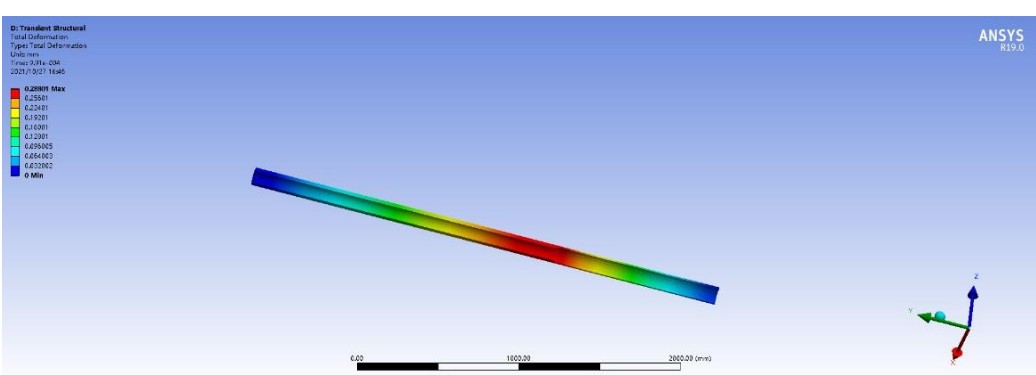

**Figure 10.** Total deformation of CrN/Cr composite coating barrel.

Figure 11 shows the maximum stress of the CrN/Cr composite coating/steel and Cr coating/steel interfaces as a function of the firing time. It can be seen that the two kinds of coating barrels show a trend of increasing first and then decreasing, which is similar to the change curve of the bore stress with the launch time, and the peak stress at the interface is also very similar. Comparing the results of Figures 8 and 11, it can be clearly seen that in the CrN/Cr composite coating barrel, the interfacial stress between the composite coating and the gun–steel matrix is reduced by 1000 MPa, which is 51% lower than the internal bore stress; however, the interface stress between the Cr coating and the gun steel substrate is only 193 MPa lower than the inner bore stress, which is only a 17% decrease, which is attributed to the higher strength and hardness of CrN/Cr [14,20], consequently greatly easing the interface stress between the coating and the substrate and achieving effective protection of the artillery barrel and the substrate.

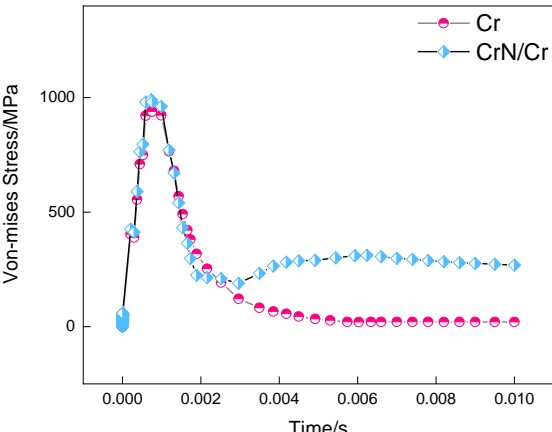

**Figure 11.** Variation curves of maximum stress at the interface between Cr coating and CrN/Cr composite coating and gun–steel matrix with gun firing time.

## 5. Conclusions

(1) With the launch of the artillery, the temperature of the inner chamber of the CrN/Cr composite coating barrel first rises, then decreases, and then stabilizes. Although the maximum temperature is higher than that of the Cr coating barrel, the temperature at the interface between the composite coating and the gun steel substrate is lower. This is because the thermal conductivity of CrN is lower than that of Cr and its specific heat capacity is higher than that of Cr.

(2) As the artillery is launched, the internal stress of the CrN/Cr composite coating barrel first increases and then decreases slightly due to the coupling of the gunpowder gas pressure and the thermal stress generated by the internal bore temperature field, and compared with the Cr coating, the interfacial stress between the composite coating and the gun–steel matrix has dropped more significantly, mainly due to the higher strength and hardness of CrN/Cr composite coating.

(3) Compared with the Cr coating, the use of CrN/Cr composite coating can greatly reduce the interface temperature between the coating and the substrate, ease the stress, and achieve effective protection of the artillery barrel. However, the CrN/Cr composite coating should focus more on the optimization of the coating at 900 mm from the bottom of the bore.

**Author Contributions:** Literature search, S.W.; simulation calculation, S.W.; data collection, S.W.; making charts, C.W.; data analysis, C.W.; finalized, W.L. All authors have read and agreed to the published version of the manuscript.

**Funding:** This work was supported by the Self-innovation Research Funding Project of Hanjiang Laboratory (HJL202012A001, HJL202012A002, HJL202012A003).

**Institutional Review Board Statement:** Not applicable.

**Informed Consent Statement:** Not applicable.

**Data Availability Statement:** Original data of this work are available upon reasonable request.

**Conflicts of Interest:** There is no conflict of interest.

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
