# Peer review of "Thermodynamic Coupling Simulation of CrN/Cr Composite Coating Barrel Bore"

_coatings, doi:10.3390/coatings11111358_

Round 1
Reviewer 1 Report
The article has the advantages of using CrN/Cr in temperature and resistance values. With that, there are few changes to be made to further improve this article.
Is there an economic advantage to using CrN/Cr?
In line 50 and 251 punctuation errors occur. You must correct these errors in the text.
I didn't find the reference number 14. However, I found this one that is interesting to be present in your work: “Dynamic stress analysis of anisotropic gun barrel under coupled thermo-mechanical loads via finite element method”.
Author Response
Thank you for your suggestion, the reply is in the word, please check

Reviewer 2 Report
The authors show an interesting article, but only describe their results, they should put the mechanisms and justify their contribution. Research on failure mechanism is essential for the prolonging of gun barrel lifetime.
Their results need to correlate with experimental (literature) results of the barrel.
The authors show an interesting article, but only describe their results, they should put the mechanisms and justify their contribution. Research on failure mechanism is essential for the prolonging of gun barrel lifetime.
Their results need to correlate with experimental (literature) results of the barrel.
The manuscript needs more details from another investigations in the field to make an introduction interesting
How original is the topic?
I think is original

Author Response

(The authors gave the same response as above.)

Reviewer 3 Report
This is the comments on the Manuscript Number: Coatings (ISSN 2079-6412) Manuscript ID
coatings-1439961
Type Article
Title Thermodynamic Coupling Simulation of CrN/Cr composite Coating Barrel Bore (Authors ShaoWei Wang, Chuanbin Wang
Rate the Manuscript:
- Significance to field and specialization of “Coatings” journal: good.
The paper contains the theoretical modeling of the thermodynamic processes in the barrel is the core component of the gun, while the inner coating is the key material to protect the barrel bore and improve its service life effectively. This paper uses the finite element method and Workbench software to simulate and analyze the temperature field and stress field changes of the Cr coating barrel and the CrN/Cr composite coating barrel during the firing of the gun. The results show that as the gun is fired, the temperature of the inner chamber of the CrN/Cr composite coating barrel first rises and then decreases and then stabilizes.
Moreover, due to the coupling effect of the gunpowder gas pressure and the thermal stress generated by the inner chamber temperature field, the inner chamber stress first rises and then decreases. It will decrease slightly after increasing. Compared with the Cr coating, although the temperature and stress of the inner chamber of the CrN/Cr composite coating barrel are higher, the interface temperature between the composite coating and the gun steel matrix is lower, and the interface stress drops more obviously, so it can greatly reduce the interface temperature and ease Stress, and then achieve effective protection of the artillery barrel.
As a results it has been established that with the launch of the artillery, the temperature of the inner chamber of the CrN/Cr composite coating barrel first rises and then decreases and then stabilizes.
Although the maximum temperature is higher than that of the Cr coating barrel, the temper ature at the interface between the composite coating and the gun steel substrate is lower. As the artillery is launched, the internal stress of the CrN/Cr composite coating barrel first increases and then decreases slightly due to the coupling of the gunpowder gas pressure and the thermal stress generated by the internal bore temperature field, and compared with the Cr coating, the interfacial stress between the composite coating and the gun-steel matrix has dropped more significantly.
Compared with Cr coating, the use of CrN/Cr composite coating can greatly reduce the interface temperature between the coating and the substrate, ease the stress, and achieve effective protection of the artillery barrel. However, the CrN/Cr composite coating should focus more on the optimization of the coating at 900mm from the bottom of the bore.
- Scientific content: good.
- Originality: good.
- Clarity and presentation: acceptable.
- Appropriateness for Journal: appropriate subject mater for the “Coatings”
- Need for rapid publication: no
Remarks
- I am repeat the abstract, because is not concise. Please, rewrite it. It must be nor similar to conclusions.
- Much more should evidence is need before achive the conclusions.
- Authors should make sure that they written every sentence to convey their meaning dearly to the not conclusion.
- Some new reference can be enrich the 14 position of very actual references.: Study of hydrogen influence on 1020 steel by low deformation method.- Materials Letters.- 184 (2016) 328–331. http://dx.doi.org/10.1016/j.matlet.2016.08.065; Analysis of electrochemical osciliations under conditions of vibration cavitation // Materials Science, 2011, vol. 47, N 1, p. 21-25. DOI: 10.1007/s11003-011-9363-z; Strength of welded joints of Cr-Mn steels with elevated content of nitrogen in hydrogen-containing media // Materials Science (Springer).– 2009, N 1, p. 97-107.; DOI: 10.1007/s11003-009-9166-7; Embrittlement of welded joints of tram rails in city environments. - Engineering Failure Analysis.- 2018. – Vol. 85, pp. 97-103. https://doi.org/10.1016/j.engfailanal.2017.12.011; Assessment of hydrogen embrittlement in high- alloy chromium-nickel steels and alloys in hydrogen at high pressures and temperatures // Strength of Materials (Springer).- 2018, vol. 50. - No 6 (456), p.880-887 https://doi.org/10.1007/s11223-019-00035-2; Thermal crack formation in TiCN/α-Al2O3 bilayer coatings grown by thermal CVD on WC-Co substrates with varied Co content; Rafael Stylianou, Dino Velic, Werner Daves, Werner Ecker, ... Christian Mitterer Article 125687; Influence of nitrided and nitrocarburised layers on the functional properties of nitrogen-doped soft carbon-based coatings deposited on 316L steel under DC glow-discharge conditions, Surface and Coatings Technology, 2020, 392, 125705.; Study of hydrogen influence on 1020 steel by low deformation method.- Materials Letters.- 184 (2016) 328–331. http://dx.doi.org/10.1016/j.matlet.2016.08.065; DOI: 10.1007/s11003-009-9166-7; Embrittlement of welded joints of tram rails in city environments. - Engineering Failure Analysis.- 2018. – Vol. 85, pp. 97-103. https://doi.org/10.1016/j.engfailanal.2017.12.011; etc.
- Recommendations: to sent after revision to journal “Coatings”.

Author Response

(The authors gave the same response as above.)

Reviewer 4 Report
The authors used the finite element method and Workbench software to simulate and analyze the behavior (temperature profile and the stress profile changes) of a Cr coating barrel and a CrN/Cr composite coating barrel, during the firing of the gun. They found some interesting advantages in using the CrN/Cr composite coating.
However, the authors must explain the aim of this study and correct all typos during the manuscript.
Some supplementary revisions are necessary:
1. At page 1, lines 25-55: The Introduction section is too short. I suggest to the authors to complete it with more information about ceramic CrN/Cr composite (e.g. structure, properties, applications, etc.) and to provide a justification of choosing the simulation methods/ software (e.g. characteristics, advantages, comparisons, etc.). Also, as I mentioned above, the authors must explain the aim of this study.
2. At page 5, lines 123-125 (Figure 1-3): The figure should be placed after its first mention in text (i.e. after actual line 140).
3. At page 5, line 142: Seems that Figure 4 is Figure 2-1. Please correct its number.
4. At page 7, lines 168-170: Please explain better the sequence of Figure 2-2 (a and b) images. Also, please use the same order of magnitude for all images (e.g. e-3). The time for the third images (2-2 a and 2-2 b) is 1.30e-3s or 1.30e-2s?
5. At page 8, lines 180-182 (Figure 2-3): The figure should be placed after its first mention in text (i.e. after actual line 183).
6. At page 8, line 185: Seems that Figure 4 is Figure 2-3. Please correct its number.
7. At page 8, line 208: Seems that Figure 3 is Figure 2-1. Please correct its number.
8. At page 10, lines 228-230 (Figure 3-3): The figure should be placed after its first mention in text (i.e. after actual line 231).
Author Response

(The authors gave the same response as above.)

Round 2
Reviewer 2 Report
the authors followed the recommendations
Reviewer 4 Report
The authors performed the suggested modifications and improved the manuscript. In these circumstances I recommend that the manuscript to be published.